# Qualitative and Quantitative Analysis of Ukrainian *Iris* Species: A Fresh Look on Their Antioxidant Content and Biological Activities

**DOI:** 10.3390/molecules25194588

**Published:** 2020-10-08

**Authors:** Olha Mykhailenko, Michal Korinek, Liudas Ivanauskas, Ivan Bezruk, Artem Myhal, Vilma Petrikaitė, Mohamed El-Shazly, Guan-Hua Lin, Chia-Yi Lin, Chia-Hung Yen, Bing-Hung Chen, Victoriya Georgiyants, Tsong-Long Hwang

**Affiliations:** 1Department of Pharmaceutical Chemistry, National University of Pharmacy, 4-Valentinivska st., 61168 Kharkiv, Ukraine; mykhailenko.farm@gmail.com (O.M.); vania.bezruk@gmail.com (I.B.); artem.migal@gmail.com (A.M.); 2Department of Biotechnology, College of Life Science, Kaohsiung Medical University, Kaohsiung 80708, Taiwan; mickorinek@hotmail.com (M.K.); bhchen@kmu.edu.tw (B.-H.C.); 3Graduate Institute of Natural Products, College of Medicine, Chang Gung University, Taoyuan 33302, Taiwan; 4Research Center for Chinese Herbal Medicine, Research Center for Food and Cosmetic Safety, and Graduate Institute of Health Industry Technology, College of Human Ecology, Chang Gung University of Science and Technology, Taoyuan 33302, Taiwan; 5Department of Analytical and Toxicological Chemistry, Lithuanian University of Health Sciences, A. Mickevičiaus g. 9, LT 44307 Kaunas, Lithuania; liudas.ivanauskas@lsmuni.lt; 6Laboratory of Drug Targets Histopathology, Institute of Cardiology, Lithuanian University of Health Sciences, Sukilėlių pr. 13, LT-50162 Kaunas, Lithuania; vilma.petrikaite@lsmuni.lt; 7Institute of Physiology and Pharmacology, Faculty of Medicine, Lithuanian University of Health Sciences, Mickeviciaus g. 9, LT-44307 Kaunas, Lithuania; 8Institute of Biotechnology, Life Sciences Centre, Vilnius University, Saulėtekio al. 7, LT-10257 Vilnius, Lithuania; 9Department of Pharmaceutical Biology, Faculty of Pharmacy and Biotechnology, the German University in Cairo, Cairo 11835, Egypt; mohamed.elshazly@pharma.asu.edu.eg; 10Department of Pharmacognosy, Faculty of Pharmacy, Ain Shams University, African Union Organization Street, Abbassia, Cairo 11566, Egypt; 11Department of Biochemistry and Molecular Biology, College of Medicine, Chang Gung University, Taoyuan 33302, Taiwan; cherrylin20170723@gmail.com (G.-H.L.); joyce950509@gmail.com (C.-Y.L.); 12Graduate Institute of Natural Products, College of Pharmacy, Kaohsiung Medical University, Kaohsiung 80708, Taiwan; chyen@kmu.edu.tw; 13Department of Medical Research, Kaohsiung Medical University Hospital, Kaohsiung 80708, Taiwan; 14The Institute of Biomedical Sciences, National Sun Yat-sen University, Kaohsiung 80424, Taiwan; 15Department of Anesthesiology, Chang Gung Memorial Hospital, Taoyuan 33305, Taiwan; 16Chinese Herbal Medicine Research Team, Healthy Aging Research Center, Chang Gung University, Taoyuan 33302, Taiwan

**Keywords:** *Iris* rhizomes, HPLC-DAD, UPLC–MS/MS, phenolic compounds, HPLC-ABTS, antioxidant, anti-inflammatory, anti-allergic, cytotoxic, coronavirus 229E

## Abstract

The major groups of antioxidant compounds (isoflavonoids, xanthones, hydroxycinnamic acids) in the rhizome methanol extracts of four Ukrainian *Iris* sp. (*Iris pallida*, *Iris hungarica*, *Iris sibirica*, and *Iris variegata*) were qualitatively and quantitatively analyzed using HPLC-DAD and UPLC-MS/MS. Gallic acid, caffeic acid, mangiferin, tectoridin, irigenin, iristectorigenin B, irisolidone, 5,6-dihydroxy-7,8,3′,5′-tetramethoxyisoflavone, irisolidone-7-*O*-*β*-d-glucopyranoside, germanaism B, and nigricin were recognized by comparing their UV/MS spectra, chromatographic retention time (tR) with those of standard reference compounds. *I. hungarica* and *I. variegata* showed the highest total amount of phenolic compounds. Germanaism B was the most abundant component in the rhizomes of *I. variegata* (7.089 ± 0.032 mg/g) and *I. hungarica* (6.285 ± 0.030 mg/g). The compound analyses showed good calibration curve linearity (r^2^ > 0.999) and low detection and quantifications limit. These results validated the method for its use in the simultaneous quantitative evaluation of phenolic compounds in the studied *Iris* sp. *I. hungarica* and *I. variegata* rhizomes exhibited antioxidant activity, as demonstrated by the HPLC-ABTS system and NRF2 expression assay and anti-inflammatory activity on respiratory burst in human neutrophils. Moreover, the extracts showed anti-allergic and cytotoxic effects against cancer cells. Anti-coronavirus 229E and lipid formation activities were also evaluated. In summary, potent antioxidant marker compounds were identified in the examined *Iris* sp.

## 1. Introduction

Oxidative stress and inflammation are pathophysiological processes that usually accompany various chronic diseases [1]. Oxidative stress and inflammatory processes are intertwined and affect each other through the activation of a plethora of molecular pathways. Oxidative stress promotes the prognosis of many diseases including inflammation, metabolic and liver diseases. The antioxidant scavenging process is an important process to prevent the harmful effect of free radicals [2]. Recently, nuclear factor erythroid 2-related factor 2 (NRF2), an antioxidant and cytoprotective factor, has received great attention because it exhibits interesting anti-inflammatory and hepatoprotective effects [3]. The uncontrolled generation of superoxide anions by human neutrophils plays a crucial role in the development of inflammatory and autoimmune disorders related to oxidative stress [4]. Thus, the use of drugs with antioxidant and/or anti-inflammatory activity is necessary to treat many oxidative stress-related diseases [5]. Most of the used antioxidants are of natural origin such as phenolics and vitamins. Phenolics are widely distributed in the plant kingdom. They can be found in many foods and medicinal plants and possess potent antioxidant properties rendering them ideal candidates for the development of antioxidant drug leads [6].

*Iris* L. (Iridaceae Juss.) is one of the largest genera of perennial herbaceous plants that comprises 1800 species [7]. *Iris* sp. are distributed in Europe, northern Africa, Asia, and the Middle East [8]. Rhizomes of various *Iris* sp. (*I. germanica* L*., I. pallida* Lam., and *I. florentina* L.) serve as a source of essential oils, which are widely used in cosmetics and perfumery [9]. The underground parts of several *Iris* sp. have been used in traditional European medicine for centuries [10]. Purified and dried rhizomes of *I. germanica*, *I. florentina,* or *I. pallida* are collectively known as *Rhizoma iridis*. They are commonly used because of their cathartic, emetic, stimulant, and expectorant properties. Dry rhizomes were used as an ingredient in tooth powders and as a chewing agent to promote teething in children. *I. germanica* is used to treat liver and spleen diseases in traditional medicinal systems [11].

Previous chemical and pharmacological studies on *Iris* sp. indicated that the plants contain several classes of secondary metabolites such as flavonoids, isoflavones, and their glycosides, *C*-glucosylxanthones, quinones, triterpenoids, and stilbene glycosides [12,13,14]. These compounds contribute to the observed immunomodulatory [15,16], estrogenic [17,18], antioxidative [19,20,21], antibacterial [22,23] and anticholinesterase [24], cytotoxic [11,25], and anti-osteoporotic activities [26]. Experimental results indicate a direct correlation between *Iris* phenolic compounds (hydroxycinnamic acids, isoflavones, flavones, xanthones) and their pharmacological activity, especially the antioxidant activity [20,27,28,29].

The global distribution of *Iris* sp. along with their potent biological activities have encouraged many research groups to study their metabolic profiles. Several reports have discussed the isolation and purification of new isoflavones from *Iris* rhizomes [25,30,31,32,33]. Scientists had to purify the new compounds using tedious chromatographic techniques and to identify the structures of the compounds using several spectroscopic techniques. These techniques are time-consuming, labor-intensive, and use excessive amounts of solvents [34,35]. Since the 1990s, new methods were developed to provide a direct approach to identify plant constituents in complex herbal extracts.

Liquid chromatography (LC) coupled with MS/MS facilitates the characterization of various compounds based on the molecular formula, exact mass, and fragmentation pattern [36,37]. HPLC-DAD-ESI-MS/MS was used for the qualitative identification of the main constituents in the rhizomes of *I. crocea, I. germanica*, and *I. spuria* from Kashmir (India) [38]. This method showed high sensitivity and allowed the identification of substances present in the raw materials in minor quantities. Sajad et al. developed an HPLC-UV-DBP method for the rapid identification and quantification of tectorigenin in *Iris* sp. growing in Kashmir [39]. The quantitative determination of tectorigenin indicated its presence in 1.08% to 8.84%. In another study, HPLC–DAD–CL and HPLC–ESI-Q-TOF-MS/MS were used for the identification of xanthones, isoflavonoid glycosides, and their aglycones, flavones, and other phenolic compounds in the rhizomes of *Belamcanda chinensis* (*I. domestica*), *I. tectorum*, and *I. dichotoma* grown in China [27]. However, most of these methods allowed the qualitative or quantitative determination of one or a few compounds but failed to present the full metabolic profile of the plants of interest.

In the last few years, several groups have used HPLC-DAD-ESI-MS^n^ for the identification of the metabolic profile of several medicinal plants [40,41,42]. This method was used by Wei et al. for the identification of known isoflavones in the rhizomes of *I. tectorum* and *I. dichotoma* grown in China. HPLC-DAD-ESI-MS^n^ can simultaneously provide UV and mass spectra, necessary for the identification of known components by comparing the chromatographic data of authentic compounds to the on-line detected chromatograms of the target compounds. It provided fragmentation pathways of the known compounds that can assist in elucidating the unknown structures based on the tandem mass [27,28,43]. Currently, HPLC coupled with several detectors is the optimal chromatographic method for the quick, simple, and quantitative identification of secondary metabolites in plant extracts [44]. Our previous phytochemical investigations on *Iris* sp. resulted in the isolation of flavones, isoflavones, xanthones, hydroxycinnamic acids, and coumarins by column chromatography. However, the qualitative and quantitative determination of phenolic compounds in certain *Iris* sp. using HPLC was never carried before. Thus, this study aimed to qualitatively and quantitatively compare the phenolic compounds in the rhizomes of four Ukrainian *Iris* sp. (*I. pallida, I. hungarica, I. sibirica*, *I. variegata*) by HPLC-DAD and UPLC-MS/MS. Furthermore, we analyzed the samples’ antioxidant capacity using the HPLC-ABTS system and NRF2 expression for the first time. We also conducted related pharmacological in vitro assays for *I. hungarica* and *I. variegata* crude extracts, including anti-inflammatory, anti-allergic, cytotoxic, hepatoprotective, and human coronavirus 229E (HCoV-229E) bioassays.

## 2. Results and Discussion 

### 2.1. Optimization of the HPLC-DAD and UPLC–MS/MS Conditions

In the current research, we applied certain modifications to an HPLC method developed for the simultaneous determination of phenolic compounds in the rhizomes of *I. dichotoma* [41]. The applied modifications resulted in a better separation of compounds with good peak symmetry. In our study, we used methanol as the extraction solvent and an ultrasonic bath to enhance the extraction efficacy. Chromatographic separation of the extracts was carried out using a Shimadzu HPLC system equipped with an ACE C_18_ column. Gradient elution was applied with 0.1% acetic acid in water-acetonitrile and acetonitrile with increasing polarity from 5% to 95%. Similar polyphenolic compounds were detected in the extracts of *I. pallida, I. hungarica, I. sibirica*, and *I. variegata* rhizomes as demonstrated by HPLC-DAD and UPLC–MS/MS analyses. Compound identification was based on their co-elution with reference compounds previously isolated from the rhizomes of *I. pseudacorus* [45] and *I. hungarica* [13,46,47], as well as based on the UV/MS spectroscopic data. For the qualitative analysis of phenolic compounds, a more selective and sensitive negative ionization mode method was selected for the crude plants [48].

### 2.2. Validation of the Methodology

The developed method was fully validated. The calibration curve, limits of detection (LOD), limits of quantification (LOQ), and the linear range for each analyte are provided in Table 1. All compounds showed good linearity (r^2^ ≥ 0.9993) within the tested ranges. The repeatability was expressed as the relative standard deviation (%RSD) of the major constituents’ content and the RSD ranged from 0.3% to 1.3%, which was satisfactory. The determination of the main compounds in the tested solutions was done by comparing the peaks retention times and the UV-spectrum obtained from the chromatogram of the standard solution (Table 2 and Appendix A). All results revealed repeatability, accuracy, high sensitivity and good linearity of the method.

### 2.3. Qualitative Analysis of the Samples

The retention times and fragmentation pattern of the investigated compounds [M − H]^−^ in the negative mode (MS^n^ spectra) were compared with the spectra of the standards. Eleven peaks were thus identified, including gallic acid (**1**), mangiferin (**2**), caffeic acid (**3**), tectoridin (**4**), germanaism B (or nigricin 4′-*O*-*β*-d-glucopyranoside) (**5**), irisolidone-7-*O*-*β*-d-glucopyranoside (**6**), iristectorigenin B (**7**), nigricin (or irisolone) (**8**), irigenin (**9**), 5,6-dihydroxy-7,8,3′,5′-tetramethoxyisoflavone (**10**), and irisolidone (**11**). All these polyphenolic compounds were qualitatively and quantitatively determined in the rhizomes of *I. pallida, I. hungaric*a, and *I. variegata*. For *I. sibirica*, only six constituents were identified, including five isoflavonoids (**4, 5**, **7**, **8**, and **9**), mangiferin (**2**), and caffeic acid (**3**). These compounds were detected in the studied *I.* species for the first time. The chromatograms of all reference standards were recorded at 269 nm (Figure 1). The sum of all major peaks area accounted for more than 90% of the total peak area in all chromatograms. The highest content of phenolic compounds was detected in the extracts of *I. variegata* and *I. hungarica* rhizomes compared with other tested *Iris* sp. (Figure 2).

Compound **1** was identified as gallic acid according to the absorbance maxima at 217 nm and 271 nm, characteristic of the hydroxycinnamic group of compounds. The presence of a molecular ion at *m/z* 169 further confirmed its nature [49]. Gallic acid is formed through the shikimic acid pathway and is a major component of many phenolic compounds [50]. Compound **3** also showed absorbance maxima at 236 nm and 324 nm corresponding to the hydroxycinnamic group of the compound. Compound **3** was eluted at t_R_ 3.92 min and showed fragment ions at *m/z* 179, 161, and 135 in the negative-ion mode, suggesting a caffeic acid structure. It is known that caffeic acid possesses potent antioxidant, anti-inflammatory, and antineoplastic properties [51,52] so its presence in the *Iris* raw materials supports their use in folk medicine targeting inflammatory-related disorders.

Compound **2** showed typical maximum absorption peaks at 240 (shoulder peak), 257, 318, and 365 nm, which are characteristic UV features of xanthones (mangiferin). Compounds **4**–**11** demonstrated maximum absorption peaks at 218–322 nm (shoulder peak) and 218–264 nm which are characteristic peaks of isoflavones (Appendix A). MS data were measured in the negative ion mode and the mass spectroscopic data of all compounds are listed in Table 3. The detected compounds demonstrated regular MS fragmentation behavior, which was useful in providing information on their chemical structures. For the flavonoid glycosides, the MS spectra showed an ion at *m/z* [(M–H) − 120]^−^ which represents a characteristic ion of *C*-glycosides, such as mangiferin (**2**). Mangiferin was the only *C*-glycosidic xanthone derivative identified in *Iris* sp. by this method. The MS spectra of flavonoid glycosides exhibited a loss of 162 Da, suggesting the presence of one hexose residue. This fragmentation pattern was characteristic of O-glycosides, such as tectoridin (**4**) and irisolidone-d-glucoside (**6**). The loss of a methyl radical ion (15 Da) was the predominant fragmentation pattern for most of the compounds, owing to the loss of a methoxy group. For example, iristectorigenin B (**7**) exhibited an ion peak at *m/z* 329 in the negative ion mode. The mass data showed a fragment ion at *m/z* 314 indicating the loss of a methyl residue. Irigenin (**9**) lost three methyl groups showing fragments at *m/z* 344 and *m/z* 329. All the LC chromatograms of the identified compounds are depicted in Figure 3. The chromatograms of the methanolic extract of *I. hungarica*, *I. variegata*, *I. pallida*, and *I. sibirica* rhizomes are illustrated in Appendix A. The pseudomolecular ion signals for germanaism B (**5**) and nigricin (**8**) were not observed in the negative ion mode utilizing the Retro-Diels-Alder (RDA) diagnostic [40], thus, for the detection of **5** and **8**, it was necessary to apply the positive ion mode [43].

In a previous report, tectoridin was identified in *I. crocea* and *I. tectorum* rhizomes by HPLC-DAD-ESI-MS/MS [38]. The presence of mangiferin and irigenin in *I. germanica* rhizomes was also demonstrated by the same authors. Also, isoflavonoids such as mangiferin, tectoridin, tectorigenin, irigenin, iristectorin A, iristectorin B, iridinirisflorentin, dichotomitin, and irilone were identified in the rhizomes of *I. dichotoma* grown in China [27]. However, the quantitative analysis of these compounds was never carried out. In the current investigation, gallic acid was only identified in *I. variegata* and *I. hungarica* rhizomes, while caffeic acid was observed in all analyzed samples.

Mangiferin is the most widespread C-glycosylxanthone in *Iris* sp. [53]. It was identified in 47 *Iris* sp. and subspecies, whereas its isomer isomangiferin was detected in 41 species [54]. Mangiferin possesses a chemotaxonomic value for *Iris* plants on the tribe, subgenus, section, and series levels. The Irideae and Tigrideae tribes may be distinguished from other Iridaceae tribes by the presence of mangiferin. In general, isoflavones were detected as the major components and could be considered as chemotaxonomic markers for these *Iris* sp.

### 2.4. Quantitative Analysis of the Samples

To estimate the potential pharmacological activities of the examined raw material, comparative quantitative analysis of each of the phenolic compounds content was carried out. The results of the HPLC quantitative analysis of the phenolic compounds in the rhizomes of each *Iris* sp. are presented in Table 4.

According to our results, *I. sibirica* rhizome extract can be distinguished from other extracts by having low amounts of phenolic compounds. The amounts of mangiferin (**2**) (0.267 ± 0.002 mg/g) and caffeic acid (**3**) (0.288 ± 0.012) were the highest among other identified compounds in this *Iris* rhizome. However, the content of all compounds including tectoridin (**4**) (0.038 ± 0.001 mg/g), germanaism B (**5**) (0.012 ± 0.000 mg/g), irisolidone-d-glucoside (**6**) (0.115 ± 0.005 mg/g), nigricin (**8**) (0.079 ± 0.002 mg/g), and irigenin (**9**) (0.069 ± 0.000 mg/g) was much lower in comparison with the other species. Compounds **1**, **7**, **10**, and **11** were absent in the extract of *I. sibirica* rhizomes which was predictable because they are considered minor metabolites of *Iris* plants.

Studies on *I. pallida* from Ukraine indicated that it does not contain a high quantity of phenolic compounds compared with other species. According to the published data [55], this species contains isoflavones irigenin, iristectorigenin A, nigricin, nigricanin, irisflorentin, iriskumaonin methyl ether, irilone, iriflogenin, and *cis*- and *trans*-α-irone. In the current investigation, high amounts of irigenin (**9**) (3.199 ± 0.034 mg/g) and tectoridin (**4**) (1.642 ± 0.023 mg/g) were detected. According to our knowledge, tectoridin, germanaism B, irisolidone-d-glucoside, iristectorigenin B, irisolidone, and 5,6-dihydroxy-7,8,3′,5′-tetramethoxyisoflavone, were identified for the first time in *I. pallida* by HPLC analysis.

The amounts of germanaism B (**5**) and irisolidone-D-glucoside (**6**) were the highest in the methanolic extracts of *I. variegata* and *I. hungarica* rhizomes (7.089 to 6.285 mg/g and 7.507 to 7.353 mg/g, respectively). The concentrations of irigenin (**9**) (5.518 ± 0.031 mg/g) and xanthone mangiferin (**2**) (5.747 ± 0.080 mg/g) in *I. variegata* were also high in comparison with the other tested *Iris* sp. According to the conducted HPLC analysis, every *Iris* sp. contained mangiferin with its amounts varying from 0.267 (*I. sibirica*) to 5.747 mg/g (*I. variegata*). These amounts were higher compared with the previous reports. For example, the amount of mangiferin in *I. dichotoma* rhizomes from different regions in China was 0.86–2.03 mg/g which was almost three times less compared with *I. variegata* from Ukraine [41]. Mangiferin has a wide range of pharmacological activities such as antiviral [56], antitumor, immunomodulating [57], antioxidant [58], and antituberculosis effects [59], thus its identification and quantification in *Iris* raw materials are important from a therapeutic perspective. Among hydroxycinnamic acids, gallic acid (**1**) was found in the extracts of *I. variegata* (3.729 ± 0.134 mg/g) and *I. hungarica* (2.362 ± 0.076 mg/g).

The most common isoflavonoid-*O*-glucosides in *I. hungarica* rhizomes were tectoridin (**4**), germanaism B (**5**), irisolodone-d-glucoside (**6**), as well as nigricin (**8**), irigenin (**9**), and irisolidone (**11**). The obtained results illustrated that the amount of tectoridin (**3**) (3.921 ± 0.071 mg/g), nigricin (**8**) (2.267 ± 0.003 mg/g) and iristectorigenin B (**7**) (0.750 ± 0.003 mg/g) in *I. hungarica* rhizomes was remarkably high in comparison with other species. However, the average content of tectoridin in *I. dichotoma* rhizomes obtained from different regions of China was reported to be 9.31 mg/g by HPLC analysis [41]. The highest amount of tectoridin (12.85 ± 0.06 mg/g) was detected in *Belamcanda chinensis* (*I. domestica*) rhizomes from Hubei Province in China [60], which significantly exceeded the content of tectoridin in *Iris* sp. from Ukraine. On the other hand, an average content of irigenin was detected in *I. domestica* from China 0.89 ± 0.08 mg/g, which was three times less than the detected amount in *I. pallida* (3.199 ± 0.034 mg/g) and *I. hungarica* (4.892 ± 0.038 mg/g) from Ukraine, and the content of irigenin in *I. variegata* exceeded five folds the reported content in *I. domestica*.

To the best of our knowledge, there was no previous report on the qualitative and quantitative determination of isoflavones such as iristectorigenin B, germanaism B, irisolidone-D-glucoside, its aglycone, nigricin, and 5,6-dihydroxy-7,8,3′,5′-tetramethoxyisoflavone in *Iris* raw materials. In a previous study, 5,6-dihydroxy-7,8,3′,5′-tetramethoxyisoflavone (**10**), a new natural compound, was isolated from *I. pseudacorus* [41]. In the current study, this compound was also identified in the other *Iris* sp. Its amount varied from 1.056 ± 0.002 mg/g in *I. hungarica*, 0.457 ± 0.003 mg/g in *I. pallida* to the highest amount (1.512 ± 0.013 mg/g) in *I. variegata*. Caffeic acid (**3**) was found in all species with the amount ranging from 0.227 to 1.515 mg/g, and the highest content was detected in *I. hungarica* rhizomes.

Out of the eleven compounds, **2**, **3**, **4**, **5**, **6**, **8**, and **9** were identified in all *Iris* sp., irisolidone-d-glucoside (**6**) was found in three species, except *I. sibirica*. The amounts of **2**, **4**, **5**, **6**, **8**, and **9** were the highest among all identified compounds in the studied *Iris* sp. Compounds **5**, **6**, **8**, and **9** were previously isolated only from *I. germanica* rhizomes [61] and were found in other *Iris* sp. [14,43,54]. These findings supported the importance of **2**, (mangiferin), **4** (tectoridin), **5** (germanaism B), **6** (irisolidone-d-glucoside), **8** (nigricin), and **9** (irigenin) as marker compounds of *Iris* sp. 

According to the results of the qualitative and quantitative analysis of the phenolic antioxidant compounds in *Iris* sp. growing in Ukraine, it can be concluded that these plants were not inferior to *Iris* sp. grown in other places around the globe. The presence and high content of phenolic compounds in *I. variegata* and *I. hungarica* encouraged us to subject these two species to intensive pharmacological investigations.

### 2.5. Pharmacology Investigation of I. variegata and I. hungarica Extracts

Phenolic compounds are known to act as antioxidants with beneficial effects on various diseases. Phenolics can prevent the development of cardiovascular diseases, cataracts, cancers, reduce fat absorption, and positively affect metabolism [62]. The potential antioxidant capacity, as well as other pharmacological activities of *Iris* rhizomes crude extracts, were evaluated in several bioassays reflecting the traditional use of *Iris* rhizomes against infection, liver, and inflammatory diseases.

#### 2.5.1. Antioxidant Activity

The HPLC-ABTS co-elution system represents a convenient method to analyze the antioxidant components in the plant crude extract [63]. The radical scavenging activities, which were expressed as Trolox equivalent antioxidant capacity (TEAC), varied among the *Iris* rhizomes water and ethanol extracts (Table 5). The antioxidant activity of *I. variegata* water extracts was the lowest (TEAC 2.92 ± 0.07 µmol/g) (Figure 4a). On the other hand, *I. hungarica* showed a potent antioxidant capacity for the water extract (TEAC 23.11 ± 0.90 µmol/g) (Figure 4b), and the ethanol extract showed the highest total antioxidant capacity (TEAC 50.32 ± 1.09 µmol/g) (Figure 4c). The antioxidant activity of the identified compounds (TEAC values, Trolox µmol/g) is displayed in Table 5. The extracts possessed antioxidant activity due to the presence of gallic acid, mangiferin, and caffeic acid. This can be explained by the fact that phenolic compounds are potent antioxidants [50,64,65] due to their high redox potential allowing them to become hydrogen donors and singlet oxygen quenchers [66]. The established antioxidant activity of the extracts was correlated with the content of the identified compounds. The higher the content of mangiferin, caffeic acid, and gallic acid, the higher the antioxidant activity (Figure 4). Higher amounts of mangiferin in *I. hungarica* together with gallic acid in the ethanolic extract accounted for more potent antioxidant capacity of the plant extract in comparison with water extracts. The obtained results were in good agreement with the previous studies [28,37].

#### 2.5.2. Anti-Inflammatory Activity of *Iris* sp. Extracts against Respiratory Burst and Degranulation by Human Neutrophils

The respiratory burst and degranulation of neutrophils are important processes in the maintenance of human health, but they need careful regulation to prevent the development of chronic and auto-immune diseases. Superoxide is a major radical produced by neutrophils and its excessive amount contributes to several acute and chronic diseases, including lung injury, sepsis, or arthritis [4]. We evaluated the effects of *Iris* extracts on superoxide anion generation and elastase release triggered by fMLF in CB-primed human neutrophils. The results revealed that the water extracts of *I. variegata* and *I. hungarica* rhizomes showed anti-inflammatory potential and inhibited superoxide anion generation at 10 μg/mL by 41.0% and 45.7%, respectively (Table 6). Interestingly, both the ethanolic and water extracts of *I. hungarica* rhizomes showed enhancing effects on elastase release by human neutrophils and thus may have immune-promoting effects related to degranulation. The observed effects of *Iris* water extracts may be correlated to the abundant isoflavone content.

#### 2.5.3. Antioxidant Capacity Expressed as NRF2 Activity

Nuclear factor erythroid 2-related factor 2 (NRF2) is a nuclear transcription factor usually activated in response to reactive oxygen species (ROS). NRF2 increases the antioxidant capability of all cells in response to stress, thus its activation is beneficial for health. It is also known that the level of NRF2 indicates the antioxidant capacity of the cells and its increase is linked with the enhanced ability to scavenge radicals [67]. Plants phenolic rich extracts were previously shown to exert a cytoprotective effect by increasing heme oxygenase-1 (HO-1) together with NRF2 [68]. In the current study, NRF2 activity was evaluated in HacaT normal skin cell line. *I. variegata* rhizomes showed a mild enhancing effect on NRF2 activity by 72.7% in normal skin cells indicating cytoprotective effects (Table 7), however, the effect did not correlate with the phenolics content (Section 2.5.1).

#### 2.5.4. Assessment of the Anti-Allergic Activity by the Inhibition of RBL-2H3 Cells Degranulation

The incidence of allergic diseases is dramatically increasing and the search for new drugs from natural sources is of great importance. We used a degranulation assay to evaluate the anti-allergic effect of *Iris* sp. To ascertain non-false positive effects of the samples that could be caused by the inhibition of cell viability, all samples were evaluated for toxicity against RBL-2H3 (rat basophilic leukemia cells) using MTT viability assay. The samples were found to be nontoxic (viability was over 96% compared with the control) at 100 μg/mL (Table 8). Samples were then evaluated for the anti-allergic activity using degranulation assay (*β*-hexosaminidase release detection assay) induced either by calcium ionophore (A23187) or antigen (anti-DNP IgE plus DNP-BSA). Calcium ionophore serves as a direct activator by facilitating calcium influx into the cell, while antigen mimics the physiological conditions of IgE-antigen complex binding to the FcεRI receptor on the mast cell membrane [69]. The results revealed that the water extract of *I. variegata* rhizomes (100 μg/mL) inhibited the degranulation of mast cells stimulated by A23187 or antigen with 38.3% and 27.0%, respectively, and the ethanolic extract of *I. hungarica* rhizomes (100 μg/mL) 22.0% and 46.7%, respectively (Table 8). Dexamethasone, a positive control, inhibited A23187- or antigen-induced *β*-hexosaminidase release by 65.7% and 66.3%, respectively.

#### 2.5.5. Cytotoxic Activity of *Iris* sp. Extracts

*I. variegata* and *I. hungarica* rhizomes aqueous extracts reduced the viability of melanoma (IGR39) (IC_50_ 0.53 and 1.15 mg/mL, respectively) and triple-negative breast cancer (MDA-MB-231) (IC_50_ 0.33 and 0.57 mg/mL, respectively) cell lines (Figure 5). *I. hungarica* rhizomes 70% ethanolic extract showed comparable efficacy to *I. variegata* water extract. Amin et al. established similar EC_50_ values for the methanolic extract of *I. kashmiriana* rhizomes from Kashmir against epithelial cancer cell lines including lung cancer A549 (IC_50_ 0.13 mg/mL) and colon cancer Caco-2: (IC_50_ 0.24 mg/mL) [70].

All extracts demonstrated lower activity against melanoma cells. Triple-negative breast cancer cells were 1.5–2 times more sensitive. It is a very interesting finding, as these cells do not possess receptors for estrogen, progesterone, and HER-2 receptors, and are usually characterized by a more aggressive nature compared with other cancer cell lines [71]. Comparing the cytotoxic effect of the aqueous and ethanolic extracts obtained from *I. hungarica* rhizomes, ethanolic extract was more effective against both melanoma (IGR39) and triple-negative breast cancer (MDA-MB-231) cells.

#### 2.5.6. Lipid Formation Activity

Non-alcoholic fatty liver disease is a common liver disease caused mainly by obesity and metabolic syndrome [72]. Lipid droplets are intracellular fat storage organelles found in most cells and are essential for all organisms. Dysregulated accumulation of lipids in cells leads to many health disorders including non-alcoholic steatohepatitis (fatty liver), obesity, type 2 diabetes, and even facilitates hepatitis type C virus infection [73]. Lipid droplets formation plays a role not only in the fatty liver but also in the process of atherosclerosis, where triacsin C, the long-chain fatty acyl CoA synthetase inhibitor, demonstrated profound effects [74]. According to our results, the water extract of *I. hungarica* rhizomes showed a 35.1% inhibitory effect on the lipid droplets in Huh7 liver cells (Table 7). 

*Iris* plants are rich in isoflavonoids and xanthones, which possess a wide range of biological activity, including anti-inflammatory, antioxidant, and antitumor properties. Phytochemical and pharmacological studies provide new insights into the possible therapeutic uses of these plants.

#### 2.5.7. Human Coronavirus 229E Activity

Human coronavirus 229E (HCoV-229E) is a strain of coronavirus family viruses, that causes upper respiratory syndrome [75]. In the screening for anti-coronavirus activity, *I. hungarica* and *I. variegata* did not show any protective effects against human coronavirus 229E (HCoV-229E) infection at 10 μg/mL (Figure 6).

## 3. Materials and Methods 

### 3.1. Chemicals and Reagents

Nine reference compounds, including mangiferin, nigricin, germanaism B, irisolidone-7-*O*-β-d-glucopyranoside, iristectorigenin B, tectoridin, irisolidone, irigenin, and 5,6-dihydroxy-7,8,3′,5′-tetramethoxyisoflavone were previously isolated from the rhizomes of *I. hungarica* and *I. pseudacorus*. The compounds were obtained by column chromatography (silica gel), identified spectroscopically and their purity was determined using UV, IR, and HPLC-MS methods. HPLC grade methanol and acetonitrile were used for the HPLC analysis. Gallic acid, and caffeic acid (purity ≥ 98.0%) (Sigma-Aldrich GmbH, Buchs, Switzerland), and HPLC grade glacial acetic acid (Fluka Chemie, Buchs, Switzerland) were used in the experiments. Other solvents and chemicals were of analytical grade.

### 3.2. Plant Materials

The rhizomes of *I. hungarica* Waldst. et Kit., *I. pallida* Lam., *I. sibirica* L. and *I. variegata* L. were obtained from the collections of M.M. Gryshko National Botanical Garden of the National Academy of Sciences of Ukraine (Kyiv, Ukraine) in October 2018. They were identified and authenticated by Dr. Buidin (Department of the Ornamental Plants, National Botanical Garden). Voucher specimens (CWN0056548, CWN0056549, CWN0056545, CWN0056534) were identified by Dr. Gamulya and were deposited at Herbarium of V.M. Karazin Kharkiv National University (Kharkiv, Ukraine). 

### 3.3. Sample Preparation 

The air-dried materials were ground to a fine powder using a laboratory mill. The powdered materials of *Iris* rhizomes (0.1 g, 60 mesh) were weighed into a volumetric flask, and methanol (10 mL) was used for extraction. The flask was placed in an ultrasonic bath at room temperature (20 ± 2 °C) for 30 min. The solutions were filtered through a membrane filter (0.45 μm) into vials made of glass. An aliquot of 20 μL was injected twice into the HPLC system for analysis. The reference compounds were used to prepare the standard solutions at a concentration of 1.0 mg/mL in methanol and were used for calibration. The samples were stored at 4 °C before use. 

### 3.4. HPLC Conditions

The separation of phenolic compounds was carried out using an ACE C_18_ column (250 mm × 4.6 mm, 5.0 μm; Zorbax Eclipse Plus, Agilent, Santa Clara, CA, USA). The flow rate of elution was 1 mL/min. The solvent system comprised solvent A (0.1% acetic acid in water) and solvent B (acetonitrile). An ultrasonic bath was used for degassing, then all solvents were filtered using a filter with a 0.22 μm membrane. A linear gradient program was applied: 0–8 min, 5–15% B; 8–30 min, 15–20% B; 30–48 min, 20–40% B; 48–58 min, 40–50% B; 58–65 min, 50%; 65–66 min, 50–95% B. The temperature of the column was constant at 25 °C. The injection volume of the sample solution was adjusted at 20 μL. The chromatograms were recorded at 269 nm (Figure 1).

### 3.5. Chromatographic Conditions for the UPLC-MS Method

Separation of the samples’ components was carried out with the ACQUITY H-class UPLC system (Waters, Milford, MA, USA) equipped with ACQUITY UPLC BEH C18 (50 × 2.1 mm, particle size 1.7 µm) (Merck Millipore, Darmstadt, Germany). Gradient elution was performed with 0.1% formic acid water solution (solvent A) and acetonitrile (solvent B), the flow rate at 0.5 mL/min. The following proportions of the solvent system were applied using a linear gradient profile B: Initial 5%, 3 min. 30%, 7 min. 50%, 7 to 8 min. 95%, 15 to 16 min. 5%. Xevo TQD triple quadrupole mass spectrometer detector (Waters) was used to obtain MS/MS data. Positive electrospray ionization was applied with the following settings: Capillary voltage was 1.5 kV, source temperature was 150 °C, desolvation temperature was 350 °C, with a desolvation gas flow 650 l/h, cone gas flow was 25 l/h. Collision energy and cone voltage were optimized for each compound separately. Collision energy varied in the range from 6eV to 20 eV and cone voltage was selected from 8 V to 38 V.

### 3.6. Identification of the Peaks and Peak Purity

The identification of the compounds **1**–**11** was achieved by HPLC analysis. The retention time (Rt), UV, MS/MS spectra of the peaks in the samples were compared with those of the authentic reference compounds. The purity of the compounds was evaluated by a diode array detector coupled with the HPLC system. The UV spectra of each peak were compared with those of the authentic reference compounds and/or by assessment of the MS/MS spectra.

### 3.7. Quantitative Determination of the Constituents

The compound concentration in the plant extract was calculated (mg/g) by the following formula:(1)$$X\;\left(\frac{\mathrm{mg}}{g\;}\right)\;=\frac{S\;\times \;\mathrm{mst}\;\times \;V}{\mathrm{Sst}\;\times \;m\;\times \;\mathrm{Vst}}$$
where S—phenolic compound peaks average area calculated from the parallel chromatograms of the sample solution; Sst—reference compound peaks average area calculated from the parallel chromatograms of the standard solution; m—powdered raw materials weights in g; mst—reference compound weights in mg; V—volumetric flask volume of the test extract in mL and Vst—volumetric flask volume of the reference compounds in mL. The results are summarized in Table 4.

### 3.8. Quantitative Analysis Validation Procedures

Following the United States Pharmacopeia (USP) recommendations, there are various analytical method validation parameters, including the limit of quantification (LOQ), the limit of detection (LOD), linearity, accuracy, and repeatability [76]. The responses’ linearity range of the standards was obtained using ten concentration levels with two injections for each level. The seven analytes were dissolved in methanol and the stock solutions were prepared. The stock solutions were diluted to a series of appropriate concentrations to construct the calibration curves. All calibration curves were recorded using the solutions of the reference compounds with an injection volume of 2.2 µL. The working solution with the lowest concentration was diluted with methanol to various concentrations. These solutions were then used for the determination of the limits of detection (LOD) and limits of quantification (LOQ) at a signal-to-noise ratio (S/N) of 3 and 10 for each compound. The repeatability was evaluated by analyzing six replicates of each preparation using HPLC (repeatability on the real sample). The main peak areas of two repeated chromatograms were used to calculate the relative standard deviation (RSD). The results are presented in Table 1 and Table 2.

### 3.9. HPLC-PDA Conditions and HPLC Post-Column Assay

HPLC-PDA and HPLC-ABTS were done using a Waters Alliance 2695 separation module system as previously described by Marksa et al. with some modifications [77]. Details are described in the Appendix A.

### 3.10. Instruments

Separation of the compounds was achieved using a Nexera X2 LC-30AD HPLC system (Shimadzu, Kyoto, Japan). The system comprises an on-line degasser, a quaternary pump, SIL-30AC autosampler (Shimadzu), CTO-20AC thermostat (Shimadzu), a column temperature controller and a SPD-M20A diode array detector (DAD). Other instruments used in the investigation were an Ultrasonic Cleaner Set (Wise Clean WUC-A06H, Witeg Labortechnik GmbH, (Wertheim Germany), Libra UniBloc AUW120D (Shimadzu Analytical Scale, Kyoto, Japan); pH-meter—Knick Electronic Battery-operated pH Meter 911 PH (Portamess, Berlin, Germany), and class A analytical vials that meet requirements of the State Pharmacopoeia of Ukraine (SPhU, 2015).

### 3.11. Extraction Procedure of Iris sp. for Bioassay

*I. variegata* and *I. hungarica* rhizomes were dried, ground, and the powder was extracted with distilled water in a water bath at 100 °C (100 g, 1 L, 60 min × 3) or 70% ethanol at room temperature (100 g, 1 L, 60 min × 3). The extracts were concentrated to dryness.

### 3.12. In-Vitro Assessment of NRF2 Activity

The activity of NRF2 reporter cells was evaluated [78]. The cell line HaCaT/ARE (antioxidant response element) was developed using a HaCaT stable cell line carrying a fragment derived from pGL4.37[luc2P/ARE/Hygro] plasmid and the luciferase reporter gene luc2P. Details are described in the Appendix A.

### 3.13. Lipid Droplet Assay

Lipid droplet assay was performed by treating Huh7 cells with BSA-conjugated oleic acid as described previously [72]. The details are described in the Appendix A.

### 3.14. Assessment of Anti-Allergic Activity Using In Vitro Assay

A methylthiazole tetrazolium (MTT) assay [79] was used to measure the possible toxic effects of the samples on RBL-2H3 cells and the experiment was performed as previously described [80]. *β*-Hexosaminidase activity assay was used to determine the degree of A23187-induced [81,82] and antigen-induced [83] degranulation in RBL-2H3 cells as previously described. The details of the assays are presented in the Appendix A.

### 3.15. Assessment of Anti-Inflammatory Activity Using In Vitro Assay

Blood was taken from healthy human donors using a protocol approved by the Chang Gung Memorial Hospital review board. Neutrophils were isolated according to the standard procedure described before [84]. The inhibition of superoxide anion generation was measured by the reduction of ferricytochrome *c* as previously described [85]. Elastase release representing the degranulation from azurophilic granules was evaluated as described before [86]. Details can be found in the Appendix A.

### 3.16. In Vitro Assessment of Cytotoxic Activity

The potential cytotoxic effect of *Iris* extracts on certain cell lines was determined by a MTT viability assay as described before [87]. Details can be found in the Appendix A.

### 3.17. Coronavirus 229E Assay

The protective effects of the samples against human coronavirus 229E (HCoV-229) was determined based on the previously described method [88]. The Huh7 cells line (human liver carcinoma cell line) was obtained from Dr. Rei-Lin Kuo (Chang Gung University, Taoyuan, Taiwan). The cells were infected with nine times the Median Tissue Culture Infectious Dose (TCID_50_) of each coronavirus 229E in the presence or absence of the compounds or vehicle. After incubation at 33 °C for 6 days, the surviving cells were then stained with MTT (3-[4.5-dimethylthiazol-2-yl]-2,5-diphenyl tetrazolium bromide). The percentage of surviving cells was then calculated.

### 3.18. Statistical Analysis

The processing of HPLC data was carried out using the LabSolutions Analysis Data System (Shimadzu). Statistical analysis was performed using one-way analysis of variance (ANOVA) followed by Tukey’s multiple comparison using Prism v.5.04 (GraphPad Software Inc., La Jolla, CA, USA, chemical composition), by Dunnet’s test (GraphPad Prism 6.0, GraphPad Software Inc., San Diego, CA, USA, anti-allergic assay), or Student’s t-test (SigmaPlot, Systat Software Inc., San Jose, CA, USA, anti-inflammatory assay). Values with *p*-values below 0.05 were considered statistically significant. The results were expressed as means ± SD (chemical analysis) or S.E.M (anti-inflammatory, anti-allergic, and antioxidant assays) values of at least three independent measurements unless otherwise specified. Two definitions were carried out in the chemical analysis.

## 4. Conclusions

In the present study, quantitative and qualitative analyses of the methanol extracts of four *Iris* sp. (*I. pallida*, *I. hungarica*, *I. sibirica*, and *I. variegata*) rhizomes were performed using a new HPLC method. Eleven phenolic compounds were identified. The identification was based on co-chromatography with reference compounds and UV/MS data. According to our analysis, mangiferin, tectoridin, germanaism B, irigenin, irisolidone-d-glucoside, and irisolidone were the major compounds of *Iris* sp. and can be proposed as chemical markers suitable for the development of quality control protocols of these species. This is the first report on the detailed analysis of the chemical composition of *I. pallida*, *I. hungarica*, *I. sibirica*, and *I. variegata*. Biological evaluation of the *Iris* sp. extracts revealed that *I. hungarica* rhizomes extract exhibited a potent antioxidant effect. The antioxidant activity was attributed to gallic acid and mangiferin content. *I. hungarica* and *I. variegata* rhizomes were for the first time shown to inhibit superoxide anion generation in fMLF-induced human neutrophils and increase the NRF2 expression. The phytochemical and pharmacological results indicated that *I. hungarica* and *I. variegata* rhizomes extract contain a balanced mixture of phenolic compounds with antioxidant, anti-inflammatory and anti-allergic biological activities.

## Figures and Tables

**Figure 1 molecules-25-04588-f001:**
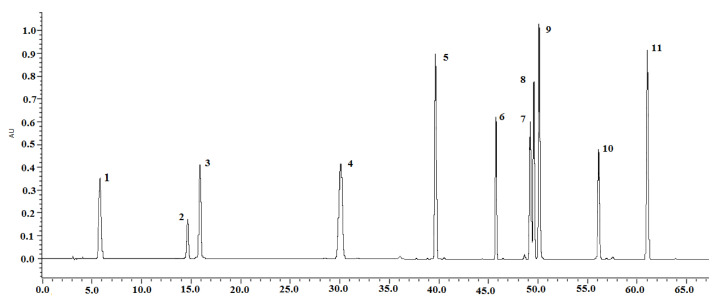
HPLC-DAD chromatograms recorded at 269 nm of the mixed reference compounds: Gallic acid (**1**), mangiferin (**2**), caffeic acid (**3**), tectoridin (**4**), germanism B (**5**), irisolidone-d-glucoside (**6**), iristectorigenin B (**7**), nigricin (**8**), irigenin (**9**), 5,6-dihydroxy-7,8,3′,5′-tetramethoxyisoflavone (**10**), and irisolidone (**11**).

**Figure 2 molecules-25-04588-f002:**
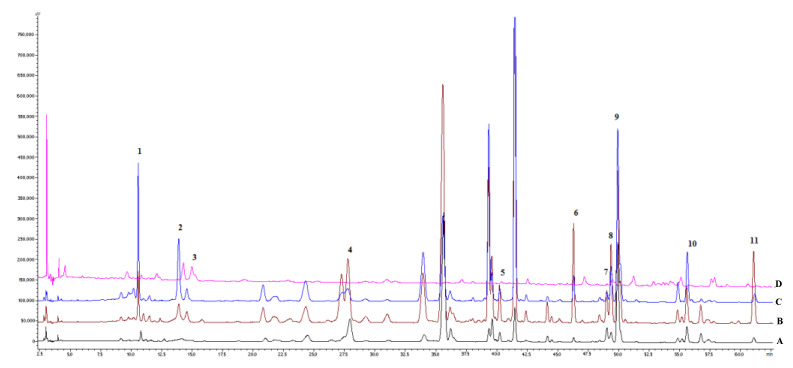
The HPLC-DAD chromatograms of the methanolic extracts of *I. pallida* (**A**), *I. hungarica* (**B**), *I. variegata* (**C**), *I. sibirica* (**D**): Gallic acid (**1**), mangiferin (**2**), caffeic acid (**3**), tectoridin (**4**), germanism B (**5**), irisolidone-d-glucoside (**6**); iristectorigenin B (**7**), nigricin (**8**), irigenin (**9**), 5,6-dihydroxy-7,8,3′,5′-tetramethoxyisoflavone (**10**), and irisolidone (**11**).

**Figure 3 molecules-25-04588-f003:**
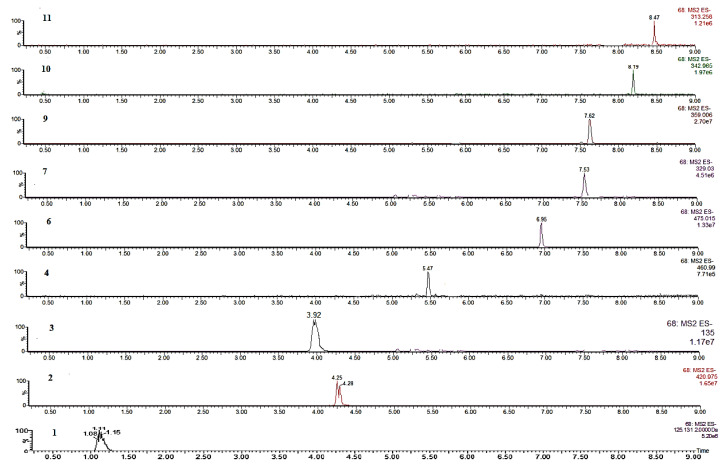
UPLC-MS chromatograms of compounds in the negative ion mode: Gallic acid (**1**) (1.14 min), mangiferin (**2**) (4.21 min), caffeic acid (**3**) (3.92 min), tectoridin (**4**) (5.47 min), irisolidone d-glucoside (**6**) (6.95 min), iristectorigenin B (**7**) (7.53 min), irigenin (**9**) (7.62 min), 5,6-dihydroxy-7,8,3′,5′-tetramethoxyisoflavone (**10**) (8.19 min), and irisolidone (**11**) (8.47 min).

**Figure 4 molecules-25-04588-f004:**
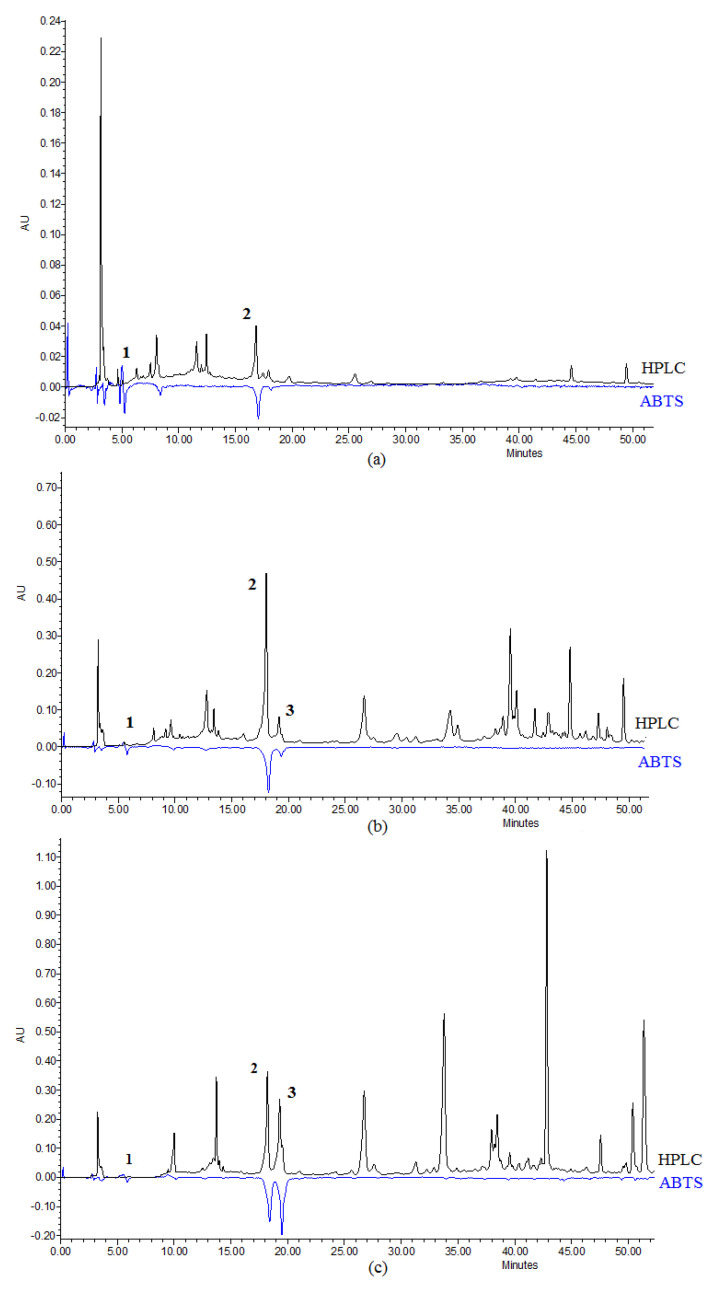
HPLC-ABTS chromatograms of (**a**) *I. variegata* rhizomes extract (H_2_O) at 247 nm (HPLC, black) and 650 nm (ABTS, blue); (**b**) *I. hungarica* rhizomes extract (H_2_O) at 255 nm/650 nm, and (**c**) *I. hungarica* rhizomes extract (70% EtOH) at 314 nm/650 nm. Gallic acid (**1**), mangiferin (**2**), and caffeic acid (**3**).

**Figure 5 molecules-25-04588-f005:**
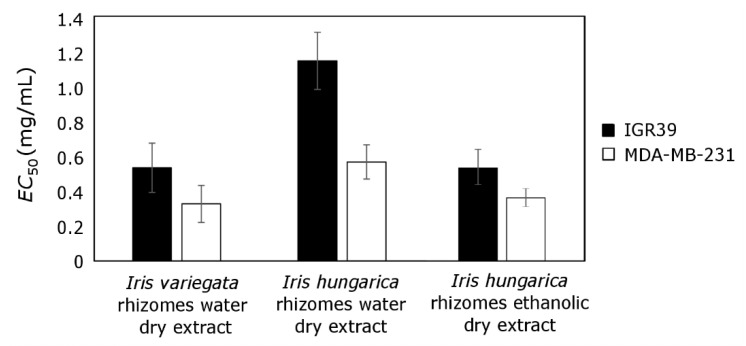
Cytotoxic effect of the tested extracts against melanoma (IGR39) and triple-negative breast cancer (MDA-MB-231) cell lines. *I. variegata* rhizomes water extract, *I. hungarica* rhizomes water extract and *I. hungarica* rhizomes ethanolic (70% EtOH) extract were tested. The values are expressed as EC_50_ values, indicating concentrations causing a 50% reduction in viability of the cells (*n* = 3).

**Figure 6 molecules-25-04588-f006:**
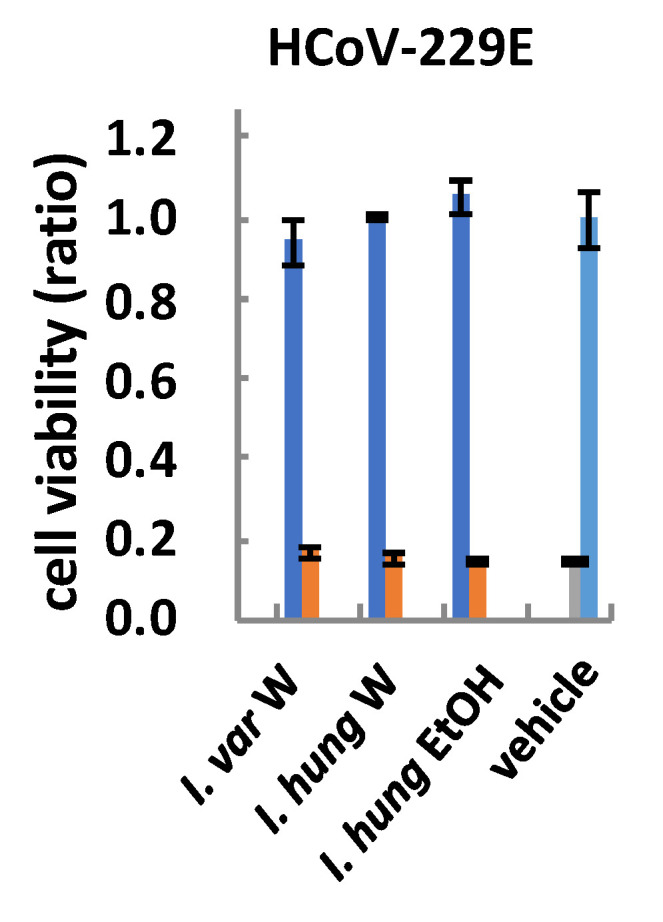
Human coronavirus 229E (HCoV-229E) protective activity of *Iris* rhizomes extracts. The cells infected by HCoV-229E were treated with the samples (orange) or vehicle (grey), any difference between them would indicate protective effects against HCoV-229E infection. The uninfected cells were also treated with the samples (dark blue) or vehicle only (light blue), serving as a control for cell viability after the treatment with the samples or vehicle. *I. var* W, *I. variegata* rhizomes (water extract); *I. hung* W, *I. hungarica* rhizomes (water extract); *I. hung* EtOH, *I. hungarica* rhizomes (ethanolic extract).

**Table 1 molecules-25-04588-t001:** Calibration curves, LOD, and LOQ data of eleven phenolic reference compounds.

Peak No	Compound	Calibration Curve ^a^	Correlation Coefficient r^2^ (*n* = 6)	Linear Range (μg/mL)	RSD (%)	LOD ^b^ (ng/mL)	LOQ ^c^ (ng/mL)
**1**	Gallic acid	y = 32880.6x − 612.983	0.9999718	0.48–61.08	1.31	30	100
**2**	Mangiferin	y = 29263.5x + 13863.9	0.9997952	0.28–145.00	1.32	310	940
**3**	Caffeic acid	y = 57646.8x − 3853.48	0.9999218	0.72–91.92	1.56	20	60
**4**	Tectoridin	y = 76104.4x + 114152	0.9995802	0.51–260.00	0.55	130	400
**5**	Germanaism B	y = 60944.8x + 123042	0.9993218	0.58–298.00	0.46	50	160
**6**	Irisolidone d-glucoside	y = 29507.2x + 5569.89	0.999981	0.49–63.1	0.98	30	90
**7**	Iristectorigenin B	y = 109562x + 68062.7	0.9996806	0.23–120.00	0.85	50	150
**8**	Nigricin	y = 89415.4x + 103288	0.9994037	0.35–181.00	0.30	40	130
**9**	Irigenin	y = 81832.6x + 137668	0.9994881	0.54–277.00	0.64	50	160
**10**	5,6-Dihydroxy-7,8,3′,5′-tetramethoxyisoflavone	y = 86268.5x + 59193.5	0.9996879	0.26–132.00	0.54	70	210
**11**	Irisolidone	y = 54297.4x + 9147.67	0.999988	0.54–69.77	1.26	10	30

^a^ compound concentration (mg/mL); y, peak area; ^b^ LOD, limit of detection (S/N = 3); ^c^ LOQ, limit of quantification (S/N = 10).

**Table 2 molecules-25-04588-t002:** Precision and stability of the eleven quantified compounds.

Peak No.	Compound	Concentration (µg/mL)	Precision				Repeatability	
			Intra-Day (*n* = 3)		Inter-Day (*n* = 3)		Recovery (%)	RSD (%)
			RSD (%)	Accuracy (%)	RSD (%)	Accuracy (%)		
**1**	Gallic acid	7.65	0.57	99.81	0.75	101.37	101.07	0.65
		30.35	0.78	99.56	0.24	102.14	99.69	0.56
		61.20	1.02	101.53	0.38	101.32	100.09	0.94
**2**	Mangiferin	9.06	0.33	100.46	0.29	100.41	100.29	0.25
		36.25	0.24	99.66	0.32	100.45	100.03	0.39
		145	0.22	100.32	1.10	98.45	99.58	0.99
**3**	Caffeic acid	11.49	1.05	102.02	0.52	98.49	100.01	0.46
		45.96	1.08	98.78	0.67	99.73	99.39	0.99
		91.92	0.64	100.35	0.95	98.17	100.17	0.37
**4**	Tectoridin	16.25	1.35	101.93	1.57	102.24	101.39	0.98
		65	1.13	101.92	0.72	101.03	100.98	0.95
		260	0.30	99.57	0.03	99.96	99.84	0.23
**5**	Germanaism B	18.62	0.65	100.92	0.16	100.23	100.38	0.48
		74.5	1.07	101.52	1.50	102.15	101.22	0.99
		298	0.64	99.09	0.93	98.69	99.26	0.68
**6**	Irisolidone-d-glucoside	0.49	1.07	98.35	0.92	101.64	100.34	1.05
		7.88	0.95	101.38	0.73	99.32	98.07	0.97
		31.55	1.02	100.44	0.94	99.78	100.74	0.31
**7**	Iristectorigenin B	7.5	1.23	101.76	1.64	102.35	101.36	0.97
		30	1.01	102.88	1.23	101.76	101.54	1.01
		120	0.07	99.90	0.33	99.53	99.81	0.25
**8**	Nigricin	11.31	1.19	101.70	1.21	101.73	101.14	0.98
		45.25	0.37	99.47	1.19	101.70	100.39	0.96
		181	0.57	99.19	0.48	99.33	99.50	0.43
**9**	Irigenin	17.31	1.08	101.54	1.29	101.84	101.12	0.98
		69.25	0.80	101.14	1.16	101.65	100.93	0.84
		277	0.33	99.53	0.20	99.71	99.74	0.24
**10**	5,6-Dihydroxy-7,8,3′,5′-tetrametoxyisoflavone	8.25	0.43	100.61	0.77	101.09	100.56	0.54
		33	0.06	100.08	0.52	100.74	100.27	0.41
		132	0.18	99.74	0.80	98.88	99.54	0.59
**11**	Irisolidone	0.54	1.07	98.74	0.52	98.24	100.06	0.52
		8.72	1.12	101.20	0.67	99.41	99.69	0.85
		34.88	0.42	100.29	0.95	100.86	100.77	0.20

**Table 3 molecules-25-04588-t003:** Chromatographic, UV, and mass spectroscopic data of the reference compounds.

Peak No	t_R_ (min)	UV *λ*max (nm)	Mol. Formula	Calculated *m/z*	Compound	[M − H]^−^ (*m*/*z*)	Fragment Ions (−)
**1**	5.96	214, 271	C_7_H_6_O_5_	170.12	Gallic acid	169	125
**2**	14.18	240, 318, 257, 365	C_19_H_18_O_11_	422.33	Mangiferin	421	403, 331, 301, 271
**3**	14.48	217, 236, 324	C_9_H_8_O_4_	180.16	Caffeic acid	179	135
**4**	29.48	263, 328	C_22_H_22_O_11_	462.41	Tectoridin	461	446, 428, 341, 299
**5**	41.08	260, 322	C_23_H_22_O_11_	474.42	Germanaism B	473	ND*
**6**	45.91	260, 330	C_23_H_23_O_11_	476.13	Irisolidone d-glucoside	475	313, 298
**7**	49.15	218, 265	C_17_H_14_O_7_	330.29	Iristectorigenin B	329	314, 311, 299, 271, 255, 164
**8**	49.50	262, 322	C_17_H_12_O_6_	312.28	Nigricin	311	ND*
**9**	50.03	264, 218	C_19_H_16_O_8_	360.32	Irigenin	359	344, 329, 314, 286, 258
**10**	56.03	222, 265	C_19_H_18_O_8_	374.35	5,6-Dihydroxy-7,8,3′,5′-tetramethoxyisoflavone	373	358, 135
**11**	61.24	259, 322	C_14_H_14_O_6_	314.08	Irisolidone	313	298

* ND—compound was not detected in the negative ion mode.

**Table 4 molecules-25-04588-t004:** Phenolic compounds content of *I. pallida*, *I. hungarica*, *I. sibirica*, and *I. variegata* rhizomes (mg/g).

Peak No	Compound	*I. pallida*	*I. hungarica*	*I. sibirica*	*I. variegata*
**1**	Gallic acid	-	2.362 ± 0.076	-	3.729 ± 0.134
**2**	Mangiferin	0.849 ± 0.029	2.368 ± 0.023	0.267 ± 0.002	5.747 ± 0.080
**3**	Caffeic acid	0.227 ± 0.033	1.515 ± 0.005	0.288 ± 0.012	1.236 ± 0.005
**4**	Tectoridin	1.642 ± 0.023	3.921 ± 0.071	0.038 ± 0.001	0.989 ± 0.006
**5**	Germanaism B	0.534 ± 0.015	6.285 ± 0.030	0.012 ± 0.000	7.089 ± 0.032
**6**	Irisolidone-d-glucoside	0.325 ± 0.030	7.353 ± 0.025	0.115 ± 0.005	7.507 ± 0.005
**7**	Iristectorigenin B	0.354 ± 0.004	0.750 ± 0.003	-	0.204 ± 0.005
**8**	Nigricin	0.317 ± 0.003	2.267 ± 0.003	0.079 ± 0.002	0.990 ± 0.010
**9**	Irigenin	3.199 ± 0.034	4.892 ± 0.038	0.069 ± 0.000	5.518 ± 0.031
**10**	5,6-Dihydroxy-7,8,3′,5′-tetramethoxyisoflavone	0.457 ± 0.003	1.056 ± 0.002	-	1.512 ± 0.013
**11**	Irisolidone	0.264 ± 0.004	4.025 ± 0.005	-	0.437 ± 0.030

Data are expressed as mean ± S.D. For each sample *n* = 2.

**Table 5 molecules-25-04588-t005:** The radical scavenging activity of individual compounds of *I. variegata* and *I. hungarica* extracts expressed as TEAC (µmol/g) using the ABTS post-column assay.

Peak No.	Component	Retention Time	*I. variegata* Rhizomes Extract (H_2_O)	*I. hungarica* Rhizomes Extract (H_2_O)	*I. hungarica* Rhizomes Extract (70% EtOH)
**1**	Gallic acid	5.78	0.52 ± 0.01	2.83 ± 0.14	3.13 ± 0.14
**2**	Mangiferin	12.68	2.40 ± 0.06	18.01 ± 0.87	20.55 ± 1.01
**3**	Caffeic acid	15.80	-	2.27 ± 0.10	26.64 ± 1.28
	Total		2.92 ± 0.07	23.11 ± 0.90	50.32 ± 1.09

**Table 6 molecules-25-04588-t006:** Anti-inflammatory activity of *Iris* sp.

Sample Description	Superoxide Anion Generation	Elastase Release
	Inh% (10 μg/mL)	Inh% (10 μg/mL)
*I. variegata* rhizomes (H_2_O)	41.0 ± 0.6 ***	13.8 ± 5.1
*I. hungarica* rhizomes (H_2_O)	45.7 ± 1.4 ***	enhancing ^a^
*I. hungarica* rhizomes (70% C_2_H_5_OH)	23.6 ± 1.3 ***	enhancing ^a^

Percentage of inhibition (Inh%) at 10 μg/mL concentration. Results are presented as mean ± S.E.M. (*n* = 3). *** *p* < 0.001 compared with the control (fMLF/CB). Genistein served as the positive control and inhibited 99.7 ± 0.6% of superoxide anion generation at 10 µg/mL and 101.2 ± 6.3% of elastase release at 30 µg/mL. ^a^ I. hungarica rhizomes (H2O, 10 μg/mL) and I. hungarica rhizomes (C2H5OH, 10 μg/mL) induced elastase release in the presence of cytochalasin B by 59.6 ± 8.1% and 42.4 ± 7.1%. Results are presented as mean ± S.E.M. (*n* = 3). Cell responses induced by fMLF/CB were expressed as 100%.

**Table 7 molecules-25-04588-t007:** Antioxidant capacity expressed as NRF2 activity and lipid droplets activity of *Iris* extracts.

Sample Description	Relative NRF2 Activity ^a^ in HacaT Cells ^b^	Lipid Droplet Inhibition Activity ^c^	
*I. variegata* rhizomes (H_2_O)	172.7	95.1 ± 11.6	
*I. hungarica* rhizomes (H_2_O)	119.9	64.9 ± 8.1	
*I. hungarica* rhizomes (70% C_2_H_5_OH)	130.8	101.5 ± 6.8	

^a^ Relative luciferase activity was calculated by normalizing luciferase activity to cell viability and is presented as the fold to solvent control. ^b^ HacaT, a normal skin cell line. The drug concentration was 100 µg/mL. TBHQ, 2-(1,1-dimethylethyl)-1,4-benzenediol (10 µM), was used as the positive control for NRF2 activation and showed 684.3 ± 37.7% of NRF2 activity. ^c^ Lipid droplet count. The average lipid droplet counts/cells of oleic acid were used as the standard representing 100% of lipid loading in Huh7 liver cell line, % mean ± S.E.M. Triacsin C (1 µM), an inhibitor of long-chain acyl-CoA synthetase, was used as the positive control and showed 16.3 ± 0.1% of lipid formation.

**Table 8 molecules-25-04588-t008:** Anti-allergic activity of *Iris* sp.

Sample Description	% Viability, RBL-2H3 ^a^	% Inhibition of A23187-Induced Degranulation ^b^		% Inhibition of Antigen-Induced Degranulation ^b^		
	100 μg/mL	10 μg/mL	100 μg/mL	10 μg/mL	100 μg/mL	
*I. variegata* rhizomes (H_2_O)	96.3 ± 0.7	12.7 ± 0.3	38.3 ± 3.5 ***	10.7 ± 3.3	27.0 ± 4.5 *	
*I. hungarica* rhizomes (H_2_O)	96.7 ± 1.7	3.7 ± 3.0	3.3 ± 2.0	4.0 ± 3.3	12.7 ± 1.7	
*I. hungarica* rhizomes (70% C_2_H_5_OH)	96.3 ± 1.9	7.3 ± 2.0	22.0 ± 5.0 *	4.7 ± 3.8	46.7 ± 2.1 ***	

^a^ The cytotoxicity of samples to RBL-2H3 was evaluated using MTT viability assay. Results are presented as mean ± S.E.M. (*n* = 3) compared with the untreated control (DMSO). Samples with viability above 85% were considered nontoxic towards RBL-2H3 cells. ^b^ Inhibition of the degranulation was assessed by A23187-induced and antigen-induced *β*-hexosaminidase release in RBL-2H3 cells. Results are presented as mean ± S.E.M. (*n* = 3); * *p* < 0.05, ** *p* < 0.01, *** *p* < 0.001 (Prism, ANOVA, Dunnet’s test) compared with the control value (A23187 or antigen only). Dexamethasone (10 nM) was used as the positive control and inhibited 65.7 ± 5.4% *** of A23187-induced and 66.3 ± 4.8% *** of antigen-induced degranulation.

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
