# Peer review of "Qualitative and Quantitative Analysis of Ukrainian Iris Species: A Fresh Look on Their Antioxidant Content and Biological Activities"

_molecules, 2020, doi:10.3390/molecules25194588_

Round 1

Reviewer 1 Report

This is an interesting study of natural components of methanolic extracts of Iris species. I have limited my comments to the analytical aspects of the submission. My main concern relates to the calibration curves and their extrapolation of detection limits (LODs) based on results from very concentrated standards. They simply need to repeat all of the curves at lower concentrations to confirm the validity of their numbers. Other specific comments follow:

Line 77: Do the author’s mean to say “distributed” rather than “distrusted”?

Line 165: It seems unusual to find a commercial 0.23 um membrane. Is there a reason not to use a more common 0.22 um pore size?

Line 169: If the authors use a diode array detector (DAD), why do they state that they only record data at 269 nm?

Line 170: Section 2.5 is labeled incorrectly as “MS parameters” as the author’s focus on the UPLC gradient and make little mention of the mass spectrometer.

Section 2.8: The authors should show representative calibration curves and a figure regarding reproducibility.

Section 2.1: It is unclear what falls within the category “Apparatus”.

Line 272: Why do the author’s define a new gradient elution that is different from section 2.5?

Line 273: The author’s point out that they identify compounds based on UV/MS data after emphasizing that they set the UV detector to 269 nm. Why not state that they use the DAD to get full UV-Vis spectra?

Line 299: Again, was data really only “recorded” at 269 nm?

Figure 2: Do these chromatograms show UV or MS/MS data? If MS/MS, the authors should specify the SRM transitions used for the detection of each compound. Was the MS operated in negative or positive ionization mode?

Table 1: Under “linear range”, please clarify the units. Is um supposed to mean “micromoles”?

Table 1: If the low end of the linear range is micromoles/mL , the extrapolated LOD and LOQ numbers are simply not valid. The author’s need to show calibration curves and they should be consistent with units (i.e., moles/volume or mass/volume). They also need to consider the adverse effects of analyte adsorption at low concentrations. Are the curves really linear or sigmoidal? In the “materials and methods” section, they should also indicate the type of sample vials (i.e., polypropylene or glass) that they use.

Table 2: Depending on the analyte, the negative effects of adsorption typically become important at low ug/mL concentrations. This table would be more valuable if the tested concentrations were reduced at least 10-fold lower.

Table 4: The reported m/z value (125) for the [M-H]- ion of gallic acid is incorrect. Also m/z 169 is not a valid fragment of gallic acid. Gallic acid is known to not fragment well by CID.

Table 4: One of the reported fragments of irisolidone D-glucoside is heavier than the reported [M-H]- ion for this molecule. In the table the authors report that the glycoside loses 162 Da but then indicate the intact glycoside molecule (m/z 475) as one of the fragments. Also, is it a glucoside (Table 4, line 343) or a glycoside (line 341)?

Table 4: Why is the parent ion of 5,6-dihydroxy-7,8,3’,5’-tetramethoxyisoflavanone 30 Da lighter than the intact molecule? Is this the loss of two methyl groups? This seems like an odd product for a low energy ESI source.

Line 323: Despite the author’s insistence that they set the UV wavelength to 269 nm, here they speak of spectra in this paragraph. Also, none of the absorbance maxima mentioned in this paragraph are at 269 nm.

Figure 3: What is the mobile phase time for this method? What is k’ for gallic acid?  Ideally it should be at least 2. The relative retention of gallic acid compared to other compounds in this figure compared to Figure 1 seems odd. Gallic acid is quite polar and rather hydrophilic. It is uncommon for this compound to be retained well on a C18 column.

I have limited knowledge regarding the rest of this submission so I will defer to the other reviewers for their critical assessment of the remaining material.

Author Response

We thank the reviewers for the positive feedback and we are grateful for the careful reading of our manuscript and for the constructive and supportive comments which we believe significantly improved the quality of our work. Please find the responses to your comments in the below-attached file. Thank you.

Reviewer 2 Report

This manuscript reports the quali-quantitative analysis of the phenolic profile of the methanol extracts obtained from the rhizomes of four Ukrainian Iris sp. namely I. pallida, I. hungarica, I. sibirica, I. variegate. Furthermore, the antioxidant capacity as well as anti-inflammatory, anti-allergic, cytotoxic, hepatoprotective and anti-coronavirus 229E (HCoV-229E) activities of I. hungarica (water and 70% ethanol extracts) and I. variegata (water extract) were evaluated.

The topic of this article is interesting and the research work was very laborious, numerous biological activities have been evaluated.

Despite this, the work presents, in my opinion, an important weakness: the phytochemical analyses and the study of biological activities were carried out on extracts prepared with different solvents. The HPLC-DAD-MSn and UPLC-MS/MS analysis on methanol extracts and the biological activity on I. hungarica (water and 70% ethanol extracts) and I. variegata water extract.

The authors should justify why they did not perform the tests related to the different activities (antioxidant, anti-inflammatory ect etc.) on the phytochemically characterized methanolic extracts... or why they did not analyse the hydroalcoholic (70% ethanol) extract or aqueous extracts which in some cases proved to be the most active (see for example antiproliferative activity ) ????

In my opinion the determinations of biological activities and the HPLC analysis should be carried out on the extract prepared with the same solvent.

Minor remark:

All the reference should be checked.

The name of the plants ( capital letter and Italics) should be corrected in the following lines 604,607, 618,621, 622, 625, 631,634, 637,639,642, 644, 646,648,651,652,655, 660,662,663,666,670,672,677,679,681,684,687,690,692,696,698,704,706,713,716,719,721,739,741,743,753, 757, 759,762, 764, 770, 774, 777, 781, 784,794,799

Author Response

We would like to thank the reviewer for the comments. Please see attached file with responses. Thank you very much

Reviewer 3 Report

It is a very extensive article, interesting both for the species under study and for its objectives.

However, I have some doubts as well as comments to suggest:

 As the authors point out, the methods used in qualitative and quantitative analysis have advantages over conventional ones. However, considering that isolation and characterized by a combination of spectroscopic techniques (DAD-HPLC, NMR, LC-MS, GC-MS, electrospray MS, tandem MS) were already used in the 90s, I would not consider them recent methods (line 101-103 and 117).

            Although these techniques have already been used to characterize some species of Iris, their application to extracts of the species considered in the work is of interest as well as the results obtained.

-Line 52-53 - I. hungarica and I. variegata showed the highest total amount of phenolic compounds.….. Was the total phenol content determined?

- The materials and methods section is very long and confusing. I suggest the convenience of summarizing it as well as reviewing some sections such as 2.5 MS parameters and 3.5 MS parameters (supplementary material) among others. I would also propose the use of chromatographic terms like Response Factor (section 2.7)

-It is not clear which extracts are used in biological tests (water extract, 70% ethanol extract) and when and why they are used.

- Perhaps Table 3 is not necessary, since that information is also shown in Table 5.

-It is not clear which extracts used in the biological assays (water extract, 70% ethanol extract) when and why they are used.

-Regarding the validation of the method - in Table 1, the peak area (y) does not appear in the corresponding equations

- Please correct the formal errors like: [7] repeated on line 81

Author Response

We thank the reviewer for the positive feedback and we are grateful for the careful reading of our manuscript and for the constructive comments which address in the attached file. Thank you very much 

Reviewer 4 Report

The paper is interesting and well written, but requires some additions. When analyzing the antioxidant potential, the authors used the HPLC-ABTS co-elution system, which is correct, but there is no data on the antioxidant activity of the whole extract, not just individual fractions. This is important as whole extract has been used for the biological activity studies. As is known, biologically active compounds can react with each other, and such interactions substantially modulate the activity of the whole extracts.

Author Response

We would like to thank the reviewer for positive feedback and we are grateful for the careful reading of our manuscript the comment. We show the total ABTS antioxidant activity of extracts in Table 6 expressed as TEAC. These values represent total antioxidant activity detected in the extract by using the HPLC-ABTS system. As shown in Figure 4, there was only a very minor antioxidant effect of other components, apart of the peak at 4 min which represents a methanol solvent peak. Thank you

Round 2

Reviewer 2 Report

In my opinion the article presents the same features and it has not be improved ...the determinations of biological activities and the HPLC analysis should be carried out on the same extracts.

Reviewer 4 Report

The paper has been improved enough. However, the technical side should be improved - the authors used the template for both Antioxidants and Molecules....